# Associations between sleep duration and insulin resistance in European children and adolescents considering the mediating role of abdominal obesity

**Barbara F. Thumann**[1,2,3], **Nathalie Michels**[2], **Regina Felső**[4], **Monica Hunsberger**[5], **Jaakko Kaprio**[6], **Luis A. Moreno**[7], **Alfonso Siani**[8], **Michael Tornaritis**[9], **Toomas Veidebaum**[10], **Stefaan De Henauw**[2], **Wolfgang Ahrens**[1,3]*, **Claudia Börnhorst**[1], on behalf of the IDEFICS and I. Family Consortia[¶]

1 Leibniz Institute for Prevention Research and Epidemiology–BIPS, Bremen, Germany, 2 Department of Public Health and Primary Care, Faculty of Medicine and Health Sciences, Ghent University, Ghent, Belgium, 3 Faculty of Mathematics and Computer Science, University of Bremen, Bremen, Germany, 4 Department of Paediatrics, Clinical Centre, University of Pécs, Pécs, Hungary, 5 Section for Epidemiology and Social Medicine (EPSO), The Sahlgrenska Academy, University of Gothenburg, Gothenburg, Sweden, 6 Department of Public Health and Institute for Molecular Medicine Finland (FIMM), University of Helsinki, Helsinki, Finland, 7 GENUD (Growth, Exercise, Nutrition and Development) Research Group, University of Zaragoza, Instituto Agroalimentario de Aragón (IA2), Instituto de Investigación Sanitaria Aragón (IIS Aragón), and Centro de Investigación Biomédica en Red Fisiopatología de la Obesidad y Nutrición (CIBERObn), Zaragoza, Spain, 8 Institute of Food Sciences, National Research Council, Avellino, Italy, 9 Research and Education Institute of Child Health, Strovolos, Cyprus, 10 Department of Chronic Diseases, National Institute for Health Development, Tallinn, Estonia

¶ Membership of the IDEFICS and I.Family Consortia is listed in the Acknowledgments.
* ahrens@leibniz-bips.de

**Data Availability Statement:** Ethical restrictions prohibit the authors from making the minimal data set publicly available because this study is based on highly sensitive data collected in young children.

## Abstract

### Background

Short sleep duration has been suggested to lead to insulin resistance both directly by altering glucose metabolism and indirectly through obesity. This study aims to investigate associations between nocturnal sleep duration and insulin resistance considering abdominal obesity as a mediator.

### Methods

We analysed data of 3 900 children aged 2–15 years participating in the second (2009/10) and third (2013/14) examination wave of the European IDEFICS/I.Family study (hereafter referred to as baseline and follow-up). Information on nocturnal sleep duration was collected by questionnaires and age-standardised (SLEEP z-score). The homeostasis model assessment (HOMA) was calculated from fasting insulin and fasting glucose obtained from blood samples; waist circumference (WAIST) was measured with an inelastic tape. HOMA and WAIST were used as indicators for insulin resistance and abdominal obesity, respectively, and transformed to age- and sex-specific z-scores. Cross-sectional and longitudinal associations between SLEEP z-score and HOMA z-score were investigated based on a path model considering WAIST z-score as a mediator adjusting for relevant confounders.

Data can only be accessed by registered scientists who are authorised to access the data with an individual account and an individual password. Statistical analyses are done on a secured central data server (CDS). It is strictly forbidden to copy or download any data from the CDS. Data are available on request and all requests need approval by the study's Steering Committee. Interested researchers can contact the IDEFICS or I.Family consortium (http://www.ideficsstudy.eu and http://www.ifamilystudy.eu) or the study coordinator (ahrens@leibniz-bips.de) to request data access. All requests for accessing data of the IDEFICS/I.Family cohort are discussed on a case-by-case basis by the Steering Committee. For this, interested parties are asked to provide details (e.g. for testing reproducibility of results) on the purpose of their request.

**Funding:** This work was done as part of the IDEFICS (http://www.idefics.eu) and I.Family studies (http://www.ifamilystudy.eu/). We gratefully acknowledge the financial support of the European Commission within the Sixth RTD Framework Program Contract No. 016181 (FOOD), and the Seventh RTD Framework Program Contract No. 266044. The publication of this article was funded by the Open Access Fund of the Leibniz Association. The funders had no role in study design, data collection and analysis, decision to publish, or preparation of the manuscript.

**Competing interests:** The authors declare no competing interests.

## Results

Cross-sectionally, baseline SLEEP z-score was negatively associated with baseline WAIST z-score (unstandardised effect estimate -0.120, 95% confidence interval [-0.167; -0.073]). We observed no direct effect of baseline SLEEP z-score on baseline HOMA z-score but a negative indirect effect through baseline WAIST z-score (-0.042 [-0.058; -0.025]). Longitudinally, there was no direct effect of baseline SLEEP z-score on HOMA z-score at follow-up but a negative indirect effect through both baseline WAIST z-score and WAIST z-score at follow-up (-0.028 [-0.040; -0.016]).

## Conclusions

Our results do not support the hypothesis of an association between short sleep duration and insulin resistance independent of abdominal obesity. However, longer sleep duration may exert short and long term beneficial effects on insulin resistance through its beneficial effects on abdominal obesity.

## Introduction

Prevalence rates of overweight and obesity in European children are at a high level although the rising trend over the last decades has been stopped in some countries [1, 2]. Obesity is associated with a higher risk for insulin resistance which is considered to be a central component of the metabolic syndrome and a risk factor for type 2 diabetes mellitus [3, 4]. Short sleep duration has been suggested as a potential risk factor for both obesity and insulin resistance [5]. Various mechanisms have been proposed to explain the links between sleep duration, obesity and insulin resistance. One hypothesis is that sleep deprivation leads to a dysregulation of appetite hormones such as leptin and ghrelin resulting in higher energy intake and weight gain [5, 6]. Apart from hormonal changes, short sleep duration might also negatively influence weight status through a behavioural pathway. For instance, more time being awake implies more time to eat [7]. Further, short sleep might cause fatigue resulting in less physical activity [7]. However, in general there is more evidence supporting the hypothesis that short sleep leads to an increase in energy intake than to a decrease in energy expenditure [7, 8]. Subsequent obesity and especially gain in visceral fat are major risk factors for insulin resistance [3]. However, short sleep duration might influence insulin resistance not only indirectly through obesity but also directly by altering glucose metabolism [5, 6].

To date, most cross-sectional and longitudinal studies conducted in young populations have found an association between short sleep duration and obesity determined by a high body mass index (BMI) [9–12]. As revealed by a review by Quist et al. [13] on sleep and cardio-metabolic risk in children and adolescents, there exists also a wealth of cross-sectional studies that investigated the association between sleep duration and waist circumference. Waist circumference has been found to be a better predictor of visceral adipose tissue than BMI [14] and is widely used as a marker of abdominal obesity. The majority of studies found longer sleep duration to be associated with lower waist circumference [13]. However, the respective longitudinal studies included in this review [13] and also one recent publication [15] showed inconsistent associations.

Both laboratory and epidemiological studies in adults have suggested short sleep duration to be a risk factor for type 2 diabetes independent of weight status [16, 17]. Also in a study in

21 adolescent boys sleep restriction over three nights was found to be associated with biological markers of insulin resistance including the homeostasis model assessment (HOMA-IR) which is calculated from fasting insulin and fasting glucose [18]. However, there are only few population-based studies that investigated the association between sleep duration and markers of insulin resistance in children and adolescents [13]. Cross-sectional studies provided only weak evidence for an association [13, 19], although two longitudinal studies reported a beneficial effect of longer sleep duration on insulin resistance [20, 21]. However, this association was independent of obesity in only one study [21]. Generally, the majority of studies in pediatric populations that investigated the association between sleep duration and insulin resistance adjusted for waist circumference or another measure of obesity, often explicitly mentioning the potential mediating role of obesity [22, 23]. However, according to our knowledge, no study to date used appropriate statistical methods to quantify this potentially mediating effect.

To close this gap, the present study aimed to investigate the potential mediating role of abdominal obesity in the cross-sectional and longitudinal associations between nocturnal sleep duration and insulin resistance.

## Subjects and methods

### Study population and procedures

The data used in this study were collected in eight European countries (Belgium, Cyprus, Estonia, Germany, Hungary, Italy, Spain and Sweden) in the framework of the IDEFICS/I.Family cohort study [24, 25]. Participants were recruited via a setting-based community-oriented approach in kindergartens and primary schools in two regions in each country; one region was defined as the intervention region, where an intervention for the prevention of childhood obesity was implemented, and the other served as the control region with no intervention [26]. The regions were selected by convenience, i.e. it was not feasible to obtain nationally representative samples [24]. However, the study followed a population-based approach and study regions were similar to the surrounding regions with respect to their socio-demographic profiles [27]. Generally, all children attending the kindergartens/primary schools in the selected regions who were between 2 and 9.9 years old were eligible to participate in the study. Further details about the study can also be inferred from the ISRCTN trial registry (http://www.isrctn.com/ISRCTN62310987).

Children were first examined between September 2007 and May 2008 (N = 16 229). Subsequently, the intervention was implemented and after its completion, 11 043 children were examined again between September 2009 and June 2010. In addition, 2 543 newly recruited children joined the IDEFICS study at this time point. For the I.Family study, children participating in IDEFICS (N = 7 117) and some of their siblings (N = 2 501) were examined again between January 2013 and June 2014. Information was collected by physical examinations, biological samples and questionnaires [25]. Usually, examinations took place on site at the kindergartens/primary schools. In addition, in some cases an examination site on the premises of the research centre, a public building or a hospital was established where examinations at kindergartens/schools were not feasible [24]. As a general procedure, questionnaires were developed in English, translated into local languages and then back-translated to maintain comparability across languages. In general, parents answered on behalf of children younger than 12 years old; older children reported for themselves. Before children entered the study, parents provided informed written consent. Moreover, children 12 years and older provided simplified written consent. Younger children gave oral consent for examinations and sample collection. Ethical approval was obtained by the appropriate Ethics Committees by each of the eight study centres conducting fieldwork, namely from the Ethics Committee of the University

Hospital Ghent (Belgium), the National Bioethics Committee of Cyprus (Cyprus), the Tallinn Medical Research Ethics Committee of the National Institutes for Health Development (Estonia), the Ethics Committee of the University of Bremen (Germany), the Scientific and Research Ethics Committee of the Medical Research Council Budapest (Hungary), the Ethics Committee of the Health Office Avellino (Italy), the Ethics Committee for Clinical Research of Aragon (Spain) and the Regional Ethical Review Board of Gothenburg (Sweden). The present study is based on the data of children and adolescents who participated in the second (2009/10) and third (2013/14) examination wave of the IDEFICS/I.Family cohort study (hereafter referred to as baseline and follow-up, respectively). This is because habitual nocturnal sleep duration was assessed for both weekdays and weekend days only during these waves.

## Waist circumference

We used waist circumference as a measure of abdominal obesity. Both at baseline and at follow-up, participants' waist circumference (cm) was measured with an inelastic tape (Seca 200, seca GmbH & Co. KG, Hamburg, Germany) in an upright position with relaxed abdomen and feet together, midway between the lowest rib margin and the iliac crest to the nearest 0.1 cm. To account for differences in waist circumference by age and sex, we calculated age- and sex-specific z-scores (WAIST z-score). For this purpose, we used percentile curves of waist circumference derived from both the IDEFICS and I.Family populations that were calculated as a function of age stratified by sex using the General Additive Model for Location Scale and Shape (GAMLSS) method [28]. Further details on the application of this method to our data can be found elsewhere [29, 30].

## Insulin resistance

Venous blood was collected from children after an overnight fast. At baseline, children who refused venepuncture were offered the alternative of giving fasting capillary blood by fingerprick. At this wave, insulin was analysed using a luminescence immunoassay (AUTO-GA Immulite 2000, Siemens, Eschborn, Germany) and blood glucose was assessed with a point-of-care analyser (Cholestech LDX, Cholestech Corp., Hayward, California, USA). At follow-up, insulin was determined by electrochemiluminescence technology (MULTI-SPOT® Assay System—Human Leptin, Insulin Assay Kit, Meso Scale Diagnostics, LLC., Rockville, Maryland, USA) and glucose with an enzymatic UV test (Cobas c701, Roche Diagnostics GmbH, Mannheim, Germany). HOMA-IR was used as a measure of insulin resistance [31] and calculated as insulin[μU/ml]*glucose[mg/dl]/405. HOMA-IR was standardised according to age and sex (HOMA z-score) using reference values derived from both the IDEFICS and I.Family populations according to previously described methods [30, 32]. As a different assessment method for insulin and glucose was used in the third examination wave, two separate reference curves were estimated for HOMA-IR depending on the assessment method used to account for the lower variation of insulin/glucose measurements in the third examination wave.

## Sleep duration

At both baseline and at follow-up, participants reported sleep duration in hours and minutes in self-completion questionnaires, i.e. the instructions read as follows: "What is the amount of time the child sleeps during a 24 hour period on weekdays? Give separate information for night time sleep and naps in the daytime." Analogously, information was collected for weekend days/vacations (in the following referred to as weekend days). We used the following formula to obtain the weighted average of nocturnal sleep duration for each child: (nocturnal sleep duration on weekdays*5 + nocturnal sleep duration on weekend days*2) / 7. The

weighted average of daily napping time (minutes) was calculated in the same manner. To account for the strong age-dependency of nocturnal sleep duration, i.e. the fact that sleep duration naturally decreases during childhood [33], we calculated an age-specific z-score (SLEEP z-score) based on the final analysis sample.

## Covariates

Covariates assessed at baseline by questionnaires included age (years), sex and highest level of parental education defined according to the "International Standard Classification of Education" (levels 0–2 = low, 3–5 = medium and 6–8 = high) [34]. A psychosocial well-being score (0–48 points with a higher score indicating higher well-being) was calculated using four subscales of the "KINDL$^R$ Questionnaire for Measuring Health-Related Quality of Life in Children and Adolescents" (emotional well-being, self-esteem, family life and relations to friends) [35]. Pubertal status was self-reported by children 8 years and older at follow-up using questions adapted from Carskadon and Acebo [36]. Girls were classified as pubertal when they reported that their menarche had already occurred and boys when they reported that their voice mutation had already started or was completed. Further, information on country of recruitment was recorded. For a sensitivity analysis, we used information on lifestyle factors collected by questionnaires at baseline. These included consumption frequencies of fruit and vegetables (times/week) as an indicator for dietary quality, time spent doing physical activity (PA) in a sports club (hours/week) as a measure of PA and the weighted average of hours of computer and TV time on weekdays and weekend days as an approximate measure of the weekly duration of electronic media use. Covariates were selected *a priori* according to pre-existing knowledge.

## Analysis dataset

As displayed in Fig 1, from the initial sample participating at both baseline and follow-up (N = 6 162), we excluded children with missing information on sleep duration and waist circumference as well as covariates (at baseline and/or follow-up) (N = 2 232). Of the remaining 3 930 participants, we excluded those with implausible values on any variable used in the analysis (N = 30), resulting in an analysis sample of 3 900 children (see Fig 1 for definitions of implausible values).

## Statistical analysis

For data preparation and descriptive analysis we used SAS 9.3 (Statistical Analysis System, SAS Institute Inc., Cary, North Carolina, USA). In order to investigate associations of interest, we conducted a path analysis in Mplus 7 [37]. With a path model, the hypothesised interrelationships among a set of variables can be investigated by modelling several related regression relationships simultaneously [37, 38]. Mediating variables in a path model are those that are a dependent variable in one relationship and an independent variable in another [37]. Hence, it is hypothesised that through a mediating variable some of the causal effects of prior variables are passed onto subsequent variables [38]. A direct effect is defined as the influence of one variable on another, unmediated by any other variable, and an indirect effect as the influence of one variable on another, mediated by at least one intervening variable [39]. The total effect is the sum of direct and indirect effect(s) of one variable on another [39]. It should be noted that the terms "direct effect", "indirect effect" and "total effect" are standard terminology in path analysis but this does not necessarily imply causality of associations. To account for non-independence of data (siblings in the sample) we used maximum likelihood estimation with robust standard errors together with the "TYPE = COMPLEX" command and the "CLUSTER"

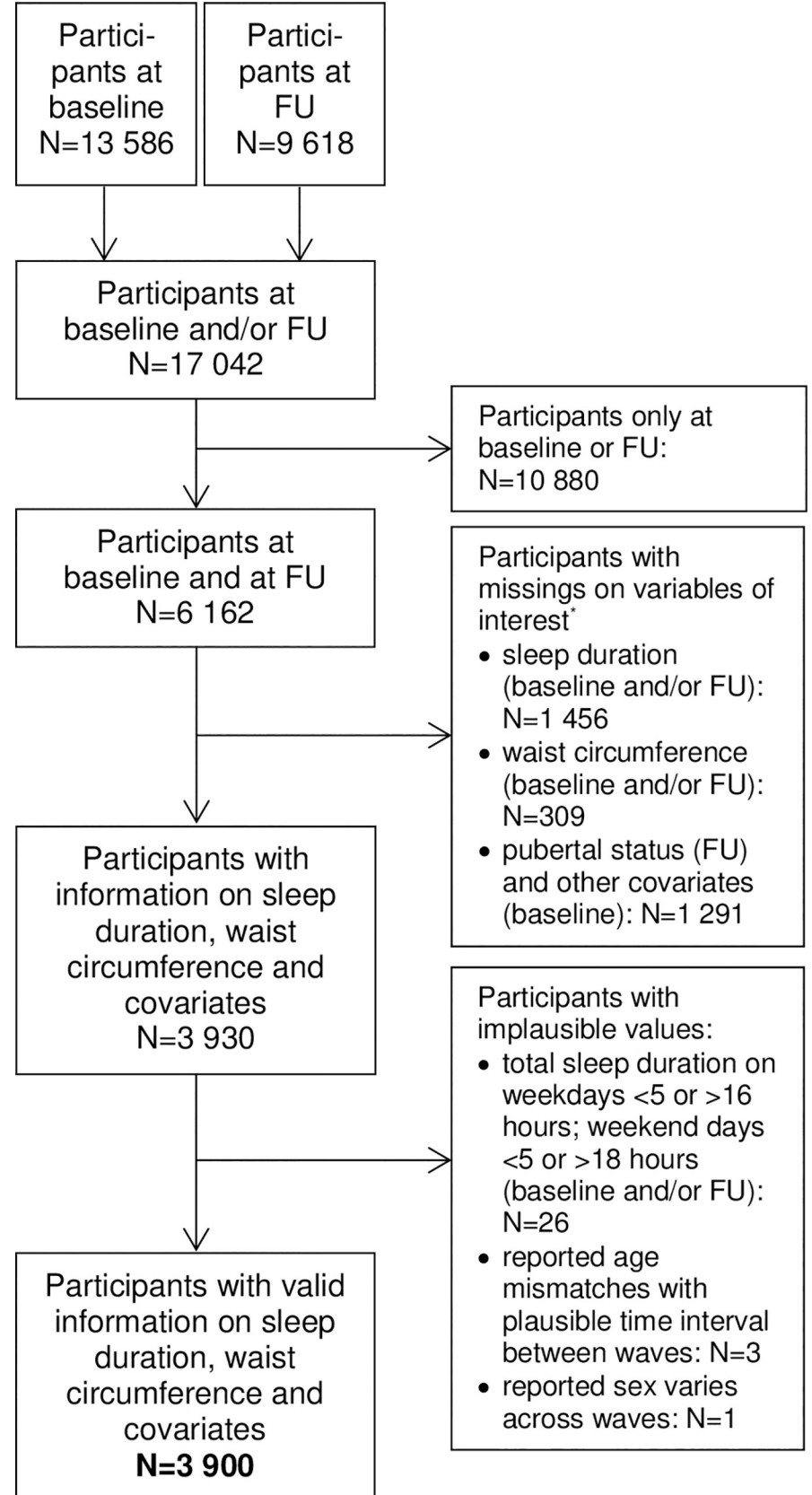

**Fig 1. Flow chart of participants, baseline examination: 2009/10; follow-up (FU) examination: 2013/14,** * **missings on multiple variables possible.**

option [37]. Maximum likelihood estimation with robust standard errors can further handle missing values in dependent variables in the model under a missing at random assumption [37]. Thus, with this technique even those participants with missing information on HOMA-IR at baseline and/or at follow-up could be included.

Guided by our conceptual framework (Fig 2), we set up a path model to investigate both cross-sectional and longitudinal associations between nocturnal sleep duration, abdominal obesity and insulin resistance within one model. Specifically, we estimated the direct effects of baseline SLEEP z-score on baseline WAIST z-score, baseline HOMA z-score, WAIST z-score at follow-up and HOMA z-score at follow-up. Further, we estimated the indirect effects of baseline SLEEP z-score on (i) baseline HOMA z-score (through baseline WAIST z-score) (ii) WAIST z-score at follow-up (through baseline WAIST z-score and SLEEP z-score at follow-up) and (iii) HOMA z-score at follow-up (through baseline WAIST z-score, baseline HOMA z-score, WAIST z-score at follow-up and SLEEP z-score at follow-up). Socio-demographic variables (baseline age, sex, country, highest educational level of parents), pubertal status at follow-up and follow-up time were included as covariates in all regressions of the path model. Furthermore, psychosocial well-being was included as a covariate because of its association with sleep duration [40] and the potential role of psychosocial stressors in the development of abdominal obesity and insulin resistance [41, 42]. Lastly, models were also adjusted for napping time because napping has been found to be associated with shorter nocturnal sleep duration [43]. The path model was also estimated stratified by age (pre-school children 2–5 years old at baseline vs. school children 6–11 years old at baseline) to examine whether there are differences between age groups.

Adequate model fit was achieved as indicated by values close to 0.95 for the Comparative Fit Index and the Tucker-Lewis Index and a value close to 0.06 for the Root Mean Square Error of Approximation (specific values see S1 Table) [44]. Multiple testing was accounted for

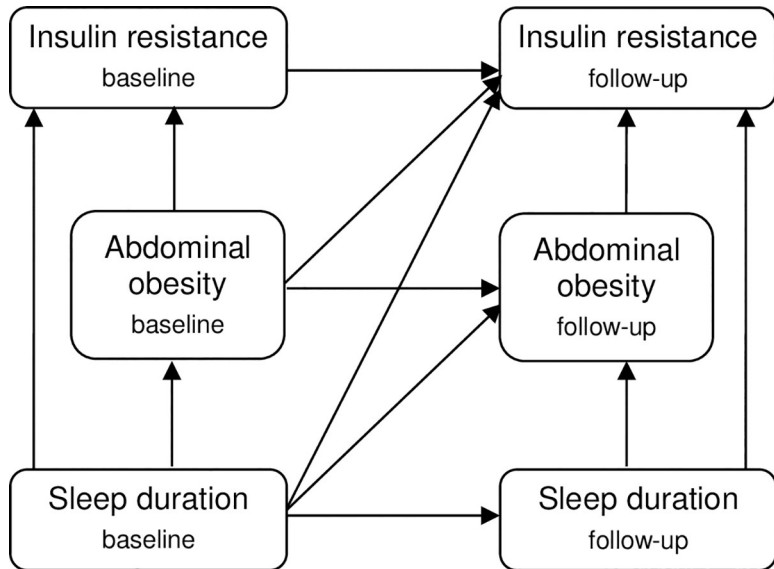

**Fig 2. Conceptual framework displaying associations assumed between sleep duration, abdominal obesity and insulin resistance at baseline and follow-up.**

by using the adjustment method of Benjamini and Hochberg [45] to control for the false discovery rate at the 0.05 level of significance resulting in an adjusted alpha level of 0.019.

Several sensitivity analyses were conducted: As it has been suggested that short sleep duration might just be a marker of an unfavourable lifestyle [10], we ran the analysis additionally adjusting all regressions of the path model for baseline consumption frequencies of fruit and vegetables, sports club physical activity and duration of electronic media use (N = 3 239). Further, as some studies indicate that children may sleep shorter during weekday nights in comparison to weekend nights [46, 47], we estimated the path model using either weekday or weekend nocturnal sleep duration as the exposure in the model (WD SLEEP z-score and WE SLEEP z-score, respectively) instead of average nocturnal sleep duration. Lastly, as the number of participants with missing values on HOMA-IR was quite high (missing only at baseline: N = 831 [21.3%], missing only at follow-up: N = 902 [23.1%], missing at both baseline and follow-up: N = 848 [21.7%]), we ran the model again with participants providing HOMA-IR at least at one time point (N = 3 052) and with participants providing complete information on all variables used in the analysis (N = 1 319).

## Results

The characteristics of the analysis group are displayed in Table 1. At baseline, the percentages of participants with a waist circumference and/or HOMA-IR value at or above the 90th age- and sex-specific reference percentile were 26.5% and 17.2%, respectively [29, 32]. Children above the 90th percentile of waist circumference and/or HOMA-IR were on average older, slept shorter at night and had less educated parents (S2 Table).

In an unadjusted analysis, baseline SLEEP z-score was negatively correlated with baseline WAIST z-score, baseline HOMA z-score and WAIST z-score at follow-up but not with HOMA z-score at follow-up (S3 Table).

Cross-sectional and longitudinal results of the path model on the association of nocturnal sleep duration with waist circumference and insulin resistance for the whole group and stratified by age are displayed in Fig 3 (direct effects) and Table 2 (indirect and total effects).

Cross-sectionally, baseline SLEEP z-score showed a negative direct effect on baseline WAIST z-score (unstandardised effect estimate -0.120; 95% confidence interval [-0.167; -0.073]) but no direct effect on baseline HOMA z-score (Fig 3). However, baseline SLEEP z-score exerted a negative indirect effect on baseline HOMA z-score through baseline WAIST z-score (-0.042 [-0.058; -0.025]), i.e. baseline HOMA z-score is expected to decrease on average by 0.042 units for every unit increase in baseline SLEEP z-score mediated through baseline WAIST z-score (Table 2). The total effect of baseline SLEEP z-score on baseline HOMA z-score was negative but did not reach statistical significance.

Longitudinally, there was no direct effect of baseline SLEEP z-score on WAIST z-score at follow-up (Fig 3) but a negative indirect effect through baseline WAIST z-score (-0.095 [-0.131; -0.058]) (Table 2). This indirect effect was the main driver of the strong total effect of baseline SLEEP z-score on WAIST z-score at follow-up (-0.108 [-0.158; -0.057]) (Table 2). Further, baseline SLEEP z-score exerted no direct effect on HOMA z-score at follow-up. However, we observed a negative indirect effect of baseline SLEEP z-score through both baseline WAIST z-score and WAIST z-score at follow-up on HOMA z-score at follow-up (-0.028 [-0.040; -0.016]) (Table 2). The total effect of baseline SLEEP z-score on HOMA z-score at follow-up was negative although not statistically significant. All other indirect effects investigated within the model were small in magnitude and mostly not statistically significant.

The age-stratified analysis (Fig 3, Table 2) showed that in both pre-school and school children effect estimates of baseline SLEEP z-score on baseline WAIST z-score and WAIST z-

**Table 1. Descriptive characteristics of the study population (N = 3 900).**

| | |
|---|---|
| *Sociodemographic information and covariates*[a] | |
| Age, mean (SD) | 7.7 (1.9) |
| Girls, N (%) | 1 980 (50.8) |
| Country, N (%) | |
| Italy | 772 (19.8) |
| Estonia | 805 (20.6) |
| Cyprus | 475 (12.2) |
| Belgium | 97 (2.5) |
| Sweden | 509 (13.1) |
| Germany | 422 (10.8) |
| Hungary | 534 (13.7) |
| Spain | 286 (7.3) |
| Highest level of parental education, N (%)[b] | |
| Low | 181 (4.6) |
| Medium | 1 639 (42.0) |
| High | 2 080 (53.3) |
| Pubertal status (pubertal), N (%) | 1 484 (38.1) |
| Well-being score, median (IQR) | 40 (36.5–43) |
| Napping (yes), N (%) | 828 (21.2) |
| Average napping time (minutes per day), median (IQR)[c] | 83 (43–103) |
| *Sleep duration, waist circumference and HOMA-IR at baseline (2009/10)* | |
| Nocturnal sleep duration (hours), mean (SD) | |
| Weekly average | 9.80 (0.78) |
| Weekday | 9.67 (0.86) |
| Weekend day | 10.12 (0.98) |
| Waist circumference (cm), median (IQR) | 56.2 (52.2–62.0) |
| Waist circumference ≥90th percentile, N (%)[d] | 1 032 (26.5) |
| HOMA-IR, median (IQR)[e] | 1.11 (0.71–1.63) |
| HOMA-IR ≥90th percentile, N (%)[d,e] | 382 (17.2) |
| *Sleep duration, waist circumference and HOMA-IR at follow-up (2013/14)* | |
| Nocturnal sleep duration (hours), mean (SD) | |
| Weekly average | 9.14 (0.93) |
| Weekday | 8.86 (1.04) |
| Weekend day | 9.86 (1.27) |
| Waist circumference (cm), median (IQR) | 64.6 (59.0–72.0) |
| Waist circumference ≥90th percentile, N (%)[d] | 1 157 (29.7) |
| HOMA-IR, median (IQR)[f] | 1.24 (0.81–1.88) |
| HOMA-IR ≥90th percentile, N (%)[d,f] | 368 (17.1) |

*HOMA-IR* homeostasis model assessment for insulin resistance, *IQR* interquartile range, *SD* standard deviation

[a] all measured at baseline (2009/10) except pubertal status which was measured at follow-up (2013/14)

[b] categorisation according to the "International Standard Classification of Education" (levels 0–2 = low, 3–5 = medium and 6–8 = high) [34]

[c] children not having a nap were not considered in this statistic (N = 3 072)

[d] based on reference values derived from data of normal weight children participating in the IDEFICS/I.Family studies according to previously described methods [29, 30, 32]

[e] missing: 1 679

[f] missing: 1 750

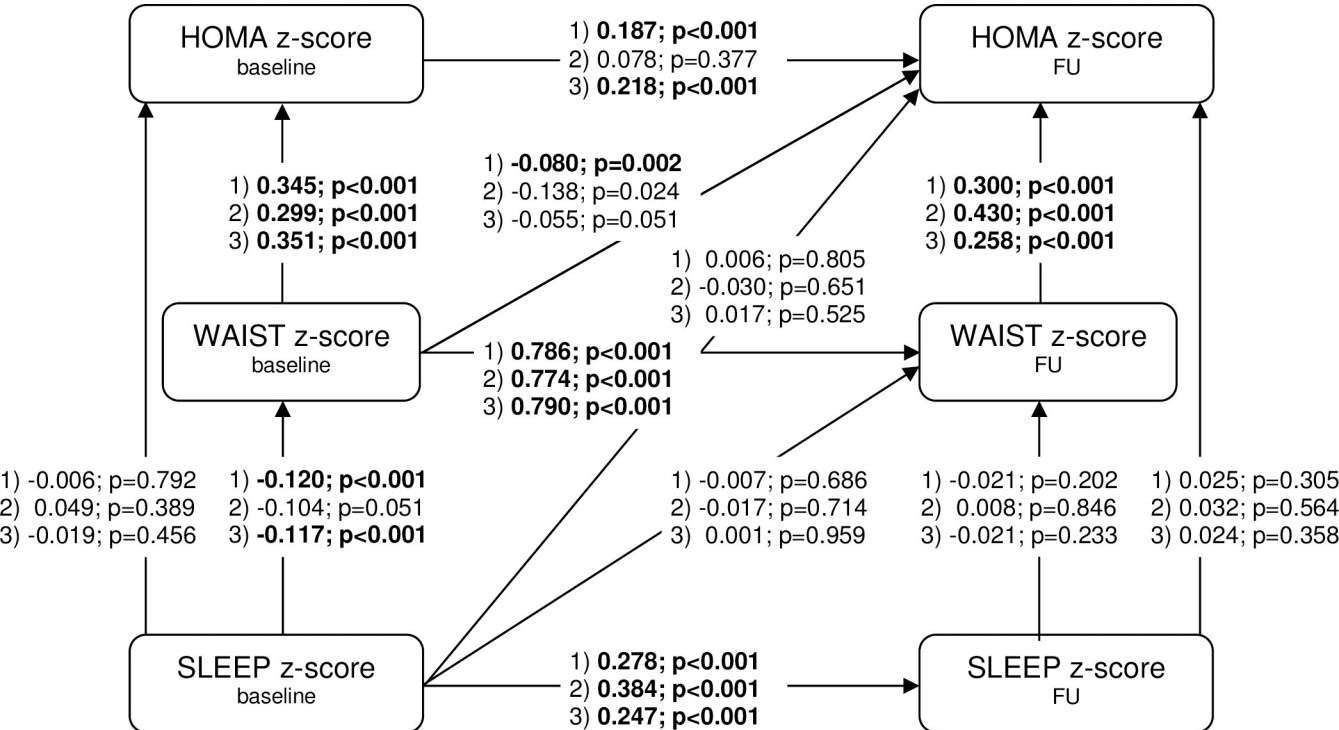

**Fig 3.** Path model for the association of nocturnal sleep duration (SLEEP) z-score with waist circumference (WAIST) z-score and homeostasis model assessment (HOMA) for insulin resistance z-score adjusted for age, sex, country, highest educational level of parents, well-being score, average napping time (all at baseline), pubertal status (at follow-up [FU]) and follow-up time: Unstandardised direct effect estimates and p-values; 1) = Whole group (N = 3 900), 2) = Pre-school children (N = 863), 3) = School children (N = 3 037); baseline: 2009/10, FU: 2013/14; bold figures indicate a false discovery rate (FDR) <0.05, an FDR adjusted significance value corresponds to $\alpha_{adj} = 0.019$.

score at follow-up generally pointed to the same direction though failing to reach statistical significance in the smaller sample of pre-school children. In school children, effect sizes of baseline SLEEP z-score on baseline HOMA z-score and HOMA z-score at follow-up were similar to those of the whole group. Overall, also the age-stratified models suggested a predominantly indirect effect of baseline SLEEP z-score on baseline HOMA z-score and HOMA z-score at follow-up through the WAIST z-score(s).

In the sensitivity analysis with additional adjustment for lifestyle factors results remained almost unchanged (S1 Fig and S4 Table). There were no major differences when distinguishing between weekday and weekend nocturnal sleep duration with the exception that there was a strong negative total effect of baseline WD SLEEP z-score on baseline HOMA z-score that was not observed for baseline WE SLEEP z-score (S2 and S3 Figs, S5 and S6 Tables). Models based on samples with varying percentages of missing values on HOMA-IR yielded similar results and did not lead to different conclusions (S4 and S5 Figs, S7 and S8 Tables).

## Discussion

To our knowledge, this is the first study quantifying the potential mediating effect of abdominal obesity in the association between nocturnal sleep duration and insulin resistance. Cross-sectionally, we found longer sleep duration to be associated with lower waist circumference and further showed sleep duration to be indirectly associated with HOMA-IR through waist circumference. In other words, our results suggest that some of the direct effect of baseline sleep duration on baseline waist circumference is transmitted onto HOMA-IR. Longer

**Table 2. Indirect and total effects and corresponding p-values obtained from path analysis of cross-sectional and longitudinal associations of nocturnal sleep duration z-score with waist circumference z-score and homeostasis model assessment for insulin resistance z-score.**

| | Whole group (N = 3 900) | | Pre-school children (N = 863) | | School children (N = 3 037) | |
|---|---|---|---|---|---|---|
| | Unst. estimate | p-value | Unst. estimate | p-value | Unst. estimate | p-value |
| *Indirect effects* | | | | | | |
| SLEEP z-score$_{baseline}$ → WAIST z-score$_{baseline}$ → HOMA z-score$_{baseline}$ | **-0.042** | **<0.001** | -0.031 | 0.053 | **-0.041** | **<0.001** |
| SLEEP z-score$_{baseline}$ → WAIST z-score$_{baseline}$ → WAIST z-score$_{FU}$ | **-0.095** | **<0.001** | -0.081 | 0.051 | **-0.092** | **<0.001** |
| SLEEP z-score$_{baseline}$ → SLEEP z-score$_{FU}$ → WAIST z-score$_{FU}$ | -0.006 | 0.204 | 0.003 | 0.846 | -0.005 | 0.237 |
| SLEEP z-score$_{baseline}$ → WAIST z-score$_{FU}$ → HOMA z-score$_{FU}$ | -0.002 | 0.687 | -0.008 | 0.715 | 0.000 | 0.959 |
| SLEEP z-score$_{baseline}$ → WAIST z-score$_{baseline}$ → HOMA z-score$_{FU}$ | **0.010** | **0.010** | 0.014 | 0.142 | 0.006 | 0.079 |
| SLEEP z-score$_{baseline}$ → HOMA z-score$_{baseline}$ → HOMA z-score$_{FU}$ | -0.001 | 0.793 | 0.004 | 0.538 | -0.004 | 0.460 |
| SLEEP z-score$_{baseline}$ → SLEEP z-score$_{FU}$ → HOMA z-score$_{FU}$ | 0.007 | 0.308 | 0.012 | 0.566 | 0.006 | 0.360 |
| SLEEP z-score$_{baseline}$ → WAIST z-score$_{baseline}$ → WAIST z-score$_{FU}$ → HOMA z-score$_{FU}$ | **-0.028** | **<0.001** | -0.035 | 0.060 | **-0.024** | **<0.001** |
| SLEEP z-score$_{baseline}$ → SLEEP z-score$_{FU}$ → WAIST z-score$_{FU}$ → HOMA z-score$_{FU}$ | -0.002 | 0.204 | 0.001 | 0.847 | -0.001 | 0.239 |
| SLEEP z-score$_{baseline}$ → WAIST z-score$_{baseline}$ → HOMA z-score$_{baseline}$ → HOMA z-score$_{FU}$ | **-0.008** | **0.001** | -0.002 | 0.431 | **-0.009** | **<0.001** |
| *Total effects* | | | | | | |
| SLEEP z-score$_{baseline}$ → HOMA z-score$_{baseline}$ | -0.048 | 0.052 | 0.018 | 0.756 | -0.060 | 0.027 |
| SLEEP z-score$_{baseline}$ → WAIST z-score$_{FU}$ | **-0.108** | **<0.001** | -0.095 | 0.130 | **-0.097** | **0.001** |
| SLEEP z-score$_{baseline}$ → HOMA z-score$_{FU}$ | -0.018 | 0.469 | -0.043 | 0.547 | -0.008 | 0.753 |

*Unst.* unstandardised; *SLEEP* nocturnal sleep duration; *WAIST* waist circumference; *HOMA* homeostasis model assessment; baseline: 2009/10, follow-up (FU): 2013/14; Path model was adjusted for age, sex, country, highest educational level of parents, well-being score, average napping time (all at baseline), pubertal status (at FU) and follow-up time; bold figures indicate a false discovery rate (FDR) <0.05, an FDR adjusted significance value corresponds to $\alpha_{adj}$ = 0.019

baseline sleep duration was also indirectly associated with lower HOMA-IR at follow-up through both baseline waist circumference and waist circumference at follow-up. This mediating role of waist circumference agrees with the well-documented link between sleep duration and abdominal obesity and the predicting role of obesity in the development of insulin resistance in childhood [4, 13, 48]. In our young study population, we did not find evidence for an association between sleep duration and HOMA-IR independent of waist circumference.

Effect sizes of the associations between nocturnal sleep duration and waist circumference and HOMA-IR, respectively, are similar to other studies using subjective measures of sleep duration [49, 50]. For instance, our cross-sectional results suggest that in comparison to a 9-year old girl who sleeps 8.5 hours/night with a waist circumference of 58.0 cm and a HOMA-IR value of 1.32, we would expect a girl of the same age who sleeps 10.2 hours/night to have a waist circumference of 56.8 cm and a HOMA-IR value of 1.23, the lower HOMA-IR value assumed to be primarily induced indirectly through the lower waist circumference. These effect sizes appear small. However, it has to be considered that sleep duration is only one among several factors potentially influencing abdominal obesity and insulin resistance in children.

Most cross-sectional studies observed similar associations between average and/or weekday sleep duration and both waist circumference and HOMA-IR as we did. A Danish study in 8–11 year old children (N = 473) found average nocturnal sleep duration to be negatively associated with both waist circumference and HOMA-IR adjusting for physical activity and sedentary time [21]. The association with HOMA-IR became non-significant after additional adjustment for fat mass index, indicating an indirect effect through fat mass [21]. A German study found total weekday sleep duration (nocturnal sleep duration and napping time

combined) to be negatively associated with waist circumference and also HOMA-IR in girls [23]. Comparably to the study by Hjorth et al. [21] and to our results, the latter association was dependent on waist circumference [23]. An association between short weekday nocturnal sleep duration and higher HOMA-IR dependent on abdominal obesity was also observed in an ethnically diverse sample of US American adolescents; the association between long week-day nocturnal sleep duration and higher HOMA-IR remained after adjustment for abdominal obesity [22]. In another study in 829 US American adolescents, objectively measured sleep duration was negatively associated with waist circumference [19]. In that study, the association of sleep duration with HOMA-IR was also negative, although not statistically significant, and was further attenuated after adjustment for BMI [19]. Contrary to our findings, in a study in 14–19 year old US American adolescents (N = 245), both average and weekday nocturnal sleep duration were found to be associated with HOMA-IR, independent of obesity indices [51]. The use of an objective method for the assessment of sleep duration but also differences in participant characteristics (e.g. inclusion of both black and white adolescents, higher proportion of overweight participants) may explain these differing results. Lastly, in contrast to our results and those of the majority of previous studies, no association between total weekday sleep duration and HOMA-IR (unadjusted for weight status) was observed in 12–17 year old adolescents participating in the European HELENA study [52].

Longitudinal studies on the associations of sleep duration with waist circumference and HOMA-IR are scarce and yielded inconsistent results. In US American children, chronic insufficient average sleep duration, determined by repeated measurements of total sleep duration from the age of 6 months to 7 years, was found to be associated with both a higher waist circumference and higher HOMA-IR at the age of 7 years (N = 652) [20]. The association with HOMA-IR became statistically non-significant after adjustment for BMI being in line with the predominantly indirect effect of baseline nocturnal sleep duration on HOMA-IR at follow-up that we observed [20]. The higher number of repeated measurements in the same subjects in that study may have resulted in a more accurate measure of habitual sleep duration enhancing the likelihood to detect also small effects. The study by Hjorth et al. [21] also included a longitudinal analysis (N = 486) showing that change in average nocturnal sleep duration over a 200-day period was not associated with change in waist circumference but with change in HOMA-IR [21]. The use of an objective method for sleep assessment, the shorter follow-up time and the different statistical analysis approach might be accountable for this discrepancy with our finding.

Strengths of our study include the large European sample and the standardised collection of questionnaire data, anthropometric measurements and biological samples in all study centres. In addition, as waist circumference and HOMA-IR are strongly dependent on age and sex, we standardised these parameters using recently developed reference values provided by the IDE-FICS/I.Family cohort study based on a huge sample of normal weight children and adolescents from all over Europe. Furthermore, the use of path analysis allowed us to explore potential short and long term effects of nocturnal sleep duration on insulin resistance in one model. However, any causal interpretation of our results relies on strong structural assumptions. For instance, we assumed baseline sleep duration to influence baseline waist circumference, although the direction of the association cannot be determined given the cross-sectional data. Further, in agreement with the literature [5, 6] and existing epidemiological studies [22, 23] we assumed abdominal obesity to mediate the association between nocturnal sleep duration and insulin resistance. Because insulin may support the onset and development of obesity [53], some researchers suggested the opposite direction, i.e. insulin resistance to mediate the association between nocturnal sleep duration and obesity [8, 9, 54]. We did not follow this hypothesis because population-based studies in children on the role of insulin resistance as a risk

factor for obesity have yielded inconsistent results [55]. Moreover, we assumed associations between sleep duration and waist circumference and HOMA-IR, respectively, to be linear. Although a few studies reported u-shaped associations [22, 56], the majority of previous studies in pediatric populations indicates that associations are more likely to be linear [13, 57]. Besides, we conducted additional analyses and did not find evidence for u-shaped associations between sleep duration and both waist circumference and HOMA-IR (S9 Table). Approximately 44% of the children in our study sample lived in the intervention regions. The intervention for the prevention of childhood obesity was implemented between 2007/08 and 2009/10. The present analysis comprised only data collected after the intervention was completed and included children from both regions. The intervention did not reveal any relevant effects on sleep duration, waist circumference and HOMA-IR [58, 59]. However, to preclude any potential effect of the intervention on our results, we conducted additional analyses adjusting the path model for residence in the intervention vs. control region as well as stratifying the model by residence in the intervention vs. control region. This did not markedly change our results (S6 and S7 Figs, S10 and S11 Tables). Another limitation of our study is that sleep duration was measured subjectively by questionnaires. Actigraphy has been described as the gold standard for sleep research under natural conditions but was not feasible in our large study for logistical and cost reasons [60]. It has been shown that sleep duration is usually overestimated when obtained by questionnaires in comparison to Actigraphs [61, 62]. Resulting non-differential measurement error of sleep duration may have led to an underestimation of effect sizes. Pubertal status was self-reported based on menarche and voice mutation. Information on breast (girls) and pubic hair (boys) development based on Tanner stages, which may be considered more accurate, was only available for a subsample (N = 2 999). However, adjustment for Tanner stage instead of menarche/voice mutation in this subsample did not alter results (S8 Fig and S12 Table). We used waist circumference as a surrogate marker for abdominal obesity that has been found to be highly correlated with visceral adipose tissue measured by magnetic resonance imaging, which is considered a gold standard measure [14]. Further, we used HOMA-IR as a surrogate marker for insulin resistance that has been validated against the hyperinsulinemic-euglycemic clamp, the gold standard measure for determining insulin resistance, and the frequently sampled intravenous glucose tolerance test, which is also considered a valid measure, with good results [63, 64]. The use of waist circumference and HOMA-IR in large-scale epidemiological studies is common and accepted because the superior gold standard measures are time consuming and burdensome for participants and hence not feasible in large population-based studies like ours, particularly not in children [4]. This choice of assessment instruments may introduce random measurement error that may attenuate the strength of any association, but this would hardly reverse the direction of any association observed. Lastly, a high proportion of children did not participate at follow-up. However, in an earlier analysis of data from the IDEFICS study overweight status, but not short nocturnal sleep duration, appeared to be a determinant of attrition [65]. Therefore, our estimates would not be biased by such a selection effect. Furthermore, the analysis method we applied allowed us to make efficient use of the existing data as even participants with missing values on some variables could be included.

## Conclusions

In conclusion, our study confirms findings of previous studies showing a cross-sectional association between longer sleep duration and lower waist circumference in a young population. The hypothesis of an association between short sleep duration and insulin resistance independent of abdominal obesity was not confirmed. However, longer sleep duration was found to be

indirectly associated with lower HOMA-IR through lower waist circumference in both cross-sectional and longitudinal analyses. Hence, our results suggest that longer nocturnal sleep duration may contribute to lower levels of insulin resistance by exerting beneficial effects on abdominal obesity.

## Supporting information

**S1 Table. Model fit indices.**
(DOCX)

**S2 Table. Descriptive characteristics of the study population at baseline by abdominal obesity and insulin resistance status at baseline.**
(DOCX)

**S3 Table. Pearson correlation coefficients and p-values among baseline SLEEP z-score, baseline WAIST z-score, baseline HOMA z-score, WAIST z-score at follow-up and HOMA z-score at follow-up.**
(DOCX)

**S4 Table. Sensitivity analysis (additional adjustment for lifestyle factors).**
(DOCX)

**S5 Table. Sensitivity analysis (weekday nocturnal sleep duration).**
(DOCX)

**S6 Table. Sensitivity analysis (weekend nocturnal sleep duration).**
(DOCX)

**S7 Table. Sensitivity analysis (complete case analysis).**
(DOCX)

**S8 Table. Sensitivity analysis (HOMA at baseline and/or follow-up).**
(DOCX)

**S9 Table. Additional analysis for investigating potential u-shaped associations.**
(DOCX)

**S10 Table. Sensitivity analysis (additional adjustment for residence in the intervention vs. control region).**
(DOCX)

**S11 Table. Sensitivity analysis (stratified by residence in the intervention vs. control region).**
(DOCX)

**S12 Table. Sensitivity analysis (adjustment for either Tanner stage or menarche/voice mutation).**
(DOCX)

**S1 Fig. Sensitivity analysis (additional adjustment for lifestyle factors).**
(DOCX)

**S2 Fig. Sensitivity analysis (weekday nocturnal sleep duration).**
(DOCX)

**S3 Fig. Sensitivity analysis (weekend nocturnal sleep duration).**
(DOCX)

**S4 Fig. Sensitivity analysis (complete case analysis).**
(DOCX)

**S5 Fig. Sensitivity analysis (HOMA at baseline and/or follow-up [FU]).**
(DOCX)

**S6 Fig. Sensitivity analysis (additional adjustment for residence in the intervention vs. control region).**
(DOCX)

**S7 Fig. Sensitivity analysis (stratified by residence in the intervention vs. control region).**
(DOCX)

**S8 Fig. Sensitivity analysis (adjustment for either Tanner stage or menarche/voice mutation).**
(DOCX)

## Acknowledgments

The authors wish to thank the IDEFICS/I.Family children and their parents for participating in the extensive examination procedures involved in this study.

## Author Contributions

**Conceptualization:** Barbara F. Thumann, Claudia Börnhorst.

**Formal analysis:** Barbara F. Thumann.

**Funding acquisition:** Wolfgang Ahrens.

**Investigation:** Nathalie Michels, Regina Felső, Monica Hunsberger, Luis A. Moreno, Alfonso Siani, Michael Tornaritis, Toomas Veidebaum, Stefaan De Henauw, Wolfgang Ahrens.

**Methodology:** Barbara F. Thumann, Jaakko Kaprio, Claudia Börnhorst.

**Supervision:** Wolfgang Ahrens, Claudia Börnhorst.

**Writing – original draft:** Barbara F. Thumann.

**Writing – review & editing:** Nathalie Michels, Regina Felső, Monica Hunsberger, Jaakko Kaprio, Luis A. Moreno, Alfonso Siani, Michael Tornaritis, Toomas Veidebaum, Stefaan De Henauw, Wolfgang Ahrens, Claudia Börnhorst.

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
