## [Decision Letter · Decision Letter 0]

15 Jan 2020

PONE-D-19-34389

Associations between sleep duration and insulin resistance in European children and adolescents considering the mediating role of abdominal obesity

PLOS ONE

Dear Dr. Ahrens,

Thank you for submitting your manuscript to PLOS ONE. After careful consideration, we feel that it has merit but does not fully meet PLOS ONE’s publication criteria as it currently stands. Therefore, we invite you to submit a revised version of the manuscript that addresses the points raised during the review process.

We would appreciate receiving your revised manuscript by Feb 29 2020 11:59PM. To enhance the reproducibility of your results, we recommend that if applicable you deposit your laboratory protocols in protocols.io, where a protocol can be assigned its own identifier (DOI) such that it can be cited independently in the future. For instructions see: http://journals.plos.org/plosone/s/submission-guidelines#loc-laboratory-protocols

We look forward to receiving your revised manuscript.

Kind regards,

Domenico Tricò, M.D.

Academic Editor

PLOS ONE

Journal Requirements:

2. In your Methods section, please provide additional information on the analysis performed. In particular, please clarify whether children who participated in the  third (2013/14) examination wave (which is called "follow-up" in the manuscript) all received the intervention. If so, please describe the intervention in the Methods section, and explain whether this was taken into consideration in your longitudinal analysis.

Moreover, please ensure you have provided sufficient details about the participant recruitment method and the demographic details of your participants,  such as: a) the recruitment date range (month and year), b) a description of any inclusion/exclusion criteria that were applied to participant recruitment, c) a table of relevant demographic details, d) a statement as to whether your sample can be considered representative of a larger population, e) a description of how participants were recruited, and f) descriptions of where participants were recruited and where the research took place.

3. Our internal editors have looked over your manuscript and determined that it is within the scope of our Determinants, Consequences and Management of Obesity  Call for Papers. This collection of papers is headed by a team of Guest Editors for PLOS ONE:Rachel Nugent and Pratibha V. Nerurkar. Additional information can be found on our announcement page: https://collections.plos.org/s/obesity-one.  

If you would like your manuscript to be considered for this collection, please let us know in your cover letter and we will ensure that your paper is treated as if you were responding to this call. If you would prefer to remove your manuscript from collection consideration, please specify this in the cover letter.

Additional Editor Comments (if provided):

Reviewers' comments:

Reviewer's Responses to Questions

**Comments to the Author**

1. Is the manuscript technically sound, and do the data support the conclusions?

Reviewer #1: Yes

Reviewer #2: Yes

2. Has the statistical analysis been performed appropriately and rigorously? 

Reviewer #1: Yes

Reviewer #2: Yes

3. Have the authors made all data underlying the findings in their manuscript fully available?

Reviewer #1: Yes

Reviewer #2: Yes

4. Is the manuscript presented in an intelligible fashion and written in standard English?

Reviewer #1: Yes

Reviewer #2: Yes

5. Review Comments to the Author

Reviewer #1: The paper by Thumann et al. analyzed the association between sleep duration and visceral obesity and insulin resistance. To this end, they longitudinally evaluated 3900 children aged 2-15 years. Waist circumference was used as expression of visceral adiposity, whereas HOMA-IR as measure of insulin resistance. Furthermore, they tested the direct and indirect effects of sleep on insulin resistance. The title and abstract are appropriate for the content of the text. Furthermore, the article is well constructed, the analysis was well performed, and the results are clearly presented.

Reviewer #2: In this study, the authors aimed to investigate the potential mediating role of abdominal obesity (waist circumference measurement) in the association between nocturnal sleep duration (measured by patient/parent report on questionnaire) and insulin resistance (HOMA-IR) in a cohort of children and adolescents. Their results do not support an association between short sleep duration and insulin resistance independent of abdominal obesity, in both cross-sectional and longitudinal analyses.

The study explores an interesting topic in a large pediatric population. Strengths of the study include (1) the large sample size and (2) the inclusion of both cross-sectional and longitudinal analyses. Also, the investigators performed appropriate sensitivity analyses.

Major Comments:

1) Analyses are based on a subjective measure of sleep duration (patient/parent report on questionnaire). This is a major limitation of the study that can affect reproducibility and the quality of the data. While given the large sample size, it would not be feasible to have performed more objective measures of sleep duration, the authors should elaborate more in the discussion section on how their measure of sleep duration may differ from the gold standard, and how this may impact results and their interpretation.

2) They use surrogate markers for insulin resistance (HOMA) and visceral fat (waist circumference). These markers although widely used are not very accurate and the authors need to discuss that.

3) The definition of puberty is not appropriate because 1) is self-reported and 2) for girls is based on menarche and for boys defined as change in voice. These phenomenon occur 2-3 years after puberty has started. Therefore, the investigators cannot really use puberty as a variable in the study neither they can draw any conclusions on the effect of puberty on the studied phenomenon.

4) The z-scores were based on their group of patients, rather than on larger healthy population.

6. PLOS authors have the option to publish the peer review history of their article (what does this mean?). If published, this will include your full peer review and any attached files.

Reviewer #1: No

Reviewer #2: No

---

## [Author Response · Author response to Decision Letter 0]

10 Mar 2020

Academic Editor: I have incorporated all of your suggestions in the revised version of the manuscript. Please find the answers to your comments in the attached file (Response to reviewers) and in the Cover Letter.

Reviewer 1: Thank you for your positive appraisal of our manuscript.

Reviewer 2: I have incorporated all of your suggestions in the revised version of the manuscript. Please find the answers to your comments in the attached file (Response to reviewers).

---

## [Decision Letter · Decision Letter 1]

14 May 2020

PONE-D-19-34389R1

Associations between sleep duration and insulin resistance in European children and adolescents considering the mediating role of abdominal obesity

PLOS ONE

Dear Ms. Thumann,

Thank you for submitting your manuscript to PLOS ONE. After careful consideration, we feel that it has merit but does not fully meet PLOS ONE’s publication criteria as it currently stands. Therefore, we invite you to submit a revised version of the manuscript that addresses the points raised during the review process.

The manuscript has been evaluated by one of the original reviewers who raised concerns in the first round of review. According to our editorial policy,  it has been also examined by a biostatistician, whose comments were overall positive and are reported below. Please address his comments by providing more information on HOMA z-score and possibly by moving some relevant data from the supplements to the main manuscript. 

We would appreciate receiving your revised manuscript by Jun 28 2020 11:59PM. To enhance the reproducibility of your results, we recommend that if applicable you deposit your laboratory protocols in protocols.io, where a protocol can be assigned its own identifier (DOI) such that it can be cited independently in the future. For instructions see: http://journals.plos.org/plosone/s/submission-guidelines#loc-laboratory-protocols

We look forward to receiving your revised manuscript.

Kind regards,

Domenico Tricò, M.D.

Academic Editor

PLOS ONE

Reviewers' comments:

Reviewer's Responses to Questions

**Comments to the Author**

1. If the authors have adequately addressed your comments raised in a previous round of review and you feel that this manuscript is now acceptable for publication, you may indicate that here to bypass the “Comments to the Author” section, enter your conflict of interest statement in the “Confidential to Editor” section, and submit your "Accept" recommendation.

Reviewer #2: All comments have been addressed

Reviewer #3: (No Response)

2. Is the manuscript technically sound, and do the data support the conclusions?

Reviewer #2: Yes

Reviewer #3: (No Response)

3. Has the statistical analysis been performed appropriately and rigorously? 

Reviewer #2: Yes

Reviewer #3: (No Response)

4. Have the authors made all data underlying the findings in their manuscript fully available?

Reviewer #2: Yes

Reviewer #3: (No Response)

5. Is the manuscript presented in an intelligible fashion and written in standard English?

Reviewer #2: Yes

Reviewer #3: (No Response)

6. Review Comments to the Author

Reviewer #2: Thank you for addressing all comments thoroughly particularly with regard to limitations of the study.

Reviewer #3: Important note: This review pertains only to ‘statistical aspects’ of the study and so ‘clinical aspects’ [like medical importance, relevance of the study, ‘clinical significance and implication(s)’ of the whole study, etc.] are to be evaluated [should be assessed] separately/independently.

Happy to note that HOMA-IR was standardised according to age and sex (HOMA z-score) [as a Biostatistician I would like you add in brief ‘how done and what effect’]. Since separate reference curves were estimated for HOMA-IR depending on the assessment method used, little more on this would have been appreciated, I guess. How about adding few diagrams (showing ‘path’ including mediating role of abdominal obesity)? [I see that there is lot of material in supplementary files, can some be in main?].

Otherwise the study is absolutely faultless and manuscript/draft of this article communicates the efforts well.

7. PLOS authors have the option to publish the peer review history of their article (what does this mean?). If published, this will include your full peer review and any attached files.

Reviewer #2: No

Reviewer #3: Yes: Dr. Sanjeev Sarmukaddam

---

## [Author Response · Author response to Decision Letter 1]

4 Jun 2020

Reviewer 3: I have incorporated all of your suggestions in the revised version of the manuscript. Please find the answers to your comments in the attached file (Response to reviewers).

---

## [Editor Report · Decision Letter 2]

9 Jun 2020

Associations between sleep duration and insulin resistance in European children and adolescents considering the mediating role of abdominal obesity

PONE-D-19-34389R2

Dear Dr. Thumann,

We’re pleased to inform you that your manuscript has been judged scientifically suitable for publication and will be formally accepted for publication once it meets all outstanding technical requirements.

Kind regards,

Domenico Tricò, M.D.

Academic Editor

PLOS ONE

---

## [Editor Report · Acceptance letter]

18 Jun 2020

PONE-D-19-34389R2 

Associations between sleep duration and insulin resistance in European children and adolescents considering the mediating role of abdominal obesity 

Dear Dr. Thumann:

I'm pleased to inform you that your manuscript has been deemed suitable for publication in PLOS ONE. Congratulations! Your manuscript is now with our production department. 

Kind regards, 

on behalf of

Dr. Domenico Tricò 

Academic Editor

PLOS ONE